# Structure-guided mutagenesis of the capsid protein indicates that a nanovirus requires assembled viral particles for systemic infection

Stefano Trapani[1]*, Eijaz Ahmed Bhat[1], Michel Yvon[2], Joséphine Lai-Kee-Him[1], François Hoh[1], Marie-Stéphanie Vernerey[2], Elodie Pirolles[2], Mélia Bonnamy[2], Guy Schoehn[3], Jean-Louis Zeddam[2], Stéphane Blanc[2]☉*, Patrick Bron[1]☉*

**1** CBS (Centre de Biologie Structurale), Univ Montpellier, CNRS, INSERM, Montpellier, France, **2** PHIM, INRAE, CIRAD, IRD, SupAgro, Univ Montpellier, Montpellier, France, **3** Univ. Grenoble Alpes, CNRS, CEA, IBS, Grenoble, France

☉ These authors contributed equally to this work.

* stefano.trapani@cbs.cnrs.fr (ST); stephane.blanc@inrae.fr (SB); patrick.bron@cbs.cnrs.fr (PB)

**Data Availability Statement:** The atomic model of the FBNSV capsid is available from the Worldwide

## Abstract

Nanoviruses are plant multipartite viruses with a genome composed of six to eight circular single-stranded DNA segments. The distinct genome segments are encapsidated individually in icosahedral particles that measure ≈18 nm in diameter. Recent studies on the model species *Faba bean necrotic stunt virus* (FBNSV) revealed that complete sets of genomic segments rarely occur in infected plant cells and that the function encoded by a given viral segment can complement the others across neighbouring cells, presumably by translocation of the gene products through unknown molecular processes. This allows the viral genome to replicate, assemble into viral particles and infect anew, even with the distinct genome segments scattered in different cells. Here, we question the form under which the FBNSV genetic material propagates long distance within the vasculature of host plants and, in particular, whether viral particle assembly is required. Using structure-guided mutagenesis based on a 3.2 Å resolution cryogenic-electron-microscopy reconstruction of the FBNSV particles, we demonstrate that specific site-directed mutations preventing capsid formation systematically suppress FBNSV long-distance movement, and thus systemic infection of host plants, despite positive detection of the mutated coat protein when the corresponding segment is agroinfiltrated into plant leaves. These results strongly suggest that the viral genome does not propagate within the plant vascular system under the form of uncoated DNA molecules or DNA:coat-protein complexes, but rather moves long distance as assembled viral particles.

## Author summary

The genome of multipartite viruses is divided in two or more segments, each encapsidated separately in an individual viral particle. An unresolved question about these viral systems

Protein Databank (http://wwpdb.org/) under the accession ID PDB-6s44 (DOI: 10.2210/pdb6S44/pdb). The cryo-EM density maps of the FBNSV are available from the Electron Microscopy Data Bank (http://www.emdatabank.org/) under the accession ID EMD-10097.

**Funding:** The CBS is a member of the French Infrastructure for Integrated Structural Biology (FRISBI), a national infrastructure supported by the French National Research Agency, ANR (grant ANR-10-INBS-05). This work was supported by Montpellier Université d'Excellence, MUSE, project BLANC-MUSE2020-Multivir, which included a post-doctoral fellowship for E.A.B. This work was supported by the French Agence Nationale de la Recherche (ANR) grant "Nanovirus" (ANR-18-CE92-0028-01). This work used the platforms of the Grenoble Instruct-ERIC centre (ISBG; UAR 3518 CNRS-CEA-UGA-EMBL) within the Grenoble Partnership for Structural Biology (PSB), supported by FRISBI (ANR-10- INBS-05-02) and GRAL, financed within the University Grenoble Alpes graduate school (Ecoles Universitaires de Recherche) CBH-EUR-GS (ANR-17-EURE-0003). The electron microscope facility is supported by the Auvergne-Rhône-Alpes Region, the Fondation Recherche Médicale (FRM), the fonds FEDER and the GIS-Infrastructures en Biologie Sante et Agronomie (IBISA). SB, MY, EP, and MSV acknowledge support from INRAE dpt. SPE, and JLZ from IRD dpt. ECOBIO. The funders had no role in study design, data collection and analysis, decision to publish, or preparation of the manuscript.

**Competing interests:** The authors have declared that no competing interests exist.

is the maintenance of the genome integrity. Two opposing hypotheses exist. One proposes that, somehow, the distinct segments can be sorted and assembled into a macromolecular complex containing at least one copy of each. This imposes either a propagation as non-encapsidated nucleic acids that could interact to form such "sorted" complexes, or structural differences between viral particles containing distinct segments also allowing assembly of sorted complexes. The other hypothesis postulates that the different segments spread independently, individually packaged in structurally similar particles, which eventually come together at random. We first determined the atomic structure of viral particles of a nanovirus (FBNSV), where no major structural differences were detected depending on the encapsidated segment. We then carried out structure-guided mutagenesis to prevent particle assembly, and thereby compromised systemic infection of host plants. These results strongly suggest that the viral genome moves long distance as assembled viral particles. Such a situation, where all particles of the viral population are alike, does not support the existence of a large macromolecular complex which would form by specifically sorting and assembling particles each containing a distinct segment. Instead, our results, together with other evidence from totally unrelated approaches, further support the independent and random propagation of the distinct segments of the FBNSV genome.

## Introduction

Member species of the genus *Nanovirus* (family *Nanoviridae*) are plant viruses which infect dicotyledonous hosts, predominantly legumes [1–3]. They possess a highly multipartite genome composed of eight circular single-stranded DNA (ssDNA) segments [4] which are individually encapsidated in separate virus particles. For all eleven nanovirus species currently recognised by the International Committee on Taxonomy of Viruses, the genomic segments (each of a size about 1 kb) contain a common stem-loop region and a single large open reading frame. Five segments, named C, M, N, R, and S, are common to all species of the genus and even of the family [1]. Segment C encodes a protein (Clink) interfering with the cell cycle. Segment M encodes the movement protein (MP). Segment N encodes the nuclear shuttle protein (NSP) that is mandatory for aphid transmission [5,6]. Segment R encodes the protein initiating replication of all the segments (M-Rep). Segment S encodes the capsid protein (CP) that assembles into virus particles, which encapsidate individually single DNA segments [7]. Three other segments, named U1, U2, and U4, are unique to the genus *Nanovirus* and encode proteins of unknown functions. Segment U4 has not been detected in one of the species of the genus [8].

How such highly multipartite viruses can be functional and have successfully evolved remains mostly unexplained [9,10]. It is particularly difficult to resolve the long-standing issue of the low probability to infect individual host cells with at least one copy of each segment, and thus the question of maintenance of the genome integrity [9]. A study on *Faba bean necrotic stunt virus* (FBNSV) [11] showed that a complete set of the eight genomic segments rarely occurs in individual cells of an infected plant. However, the function encoded by a given segment can complement the others at a distance, in cells where this segment is absent. It was proposed that this so-called pluri-cellular way of life is allowed by translocation of the gene products (e.g. mRNA, proteins, or both) among distinct infected cells, through molecular mechanisms that remain elusive. This phenomenon thus drives the viral genome to replicate, self-assemble into viral particles and infect anew even with the distinct segments scattered in distinct cells.

An intriguing question then arises: how and under what form do the genome segments of FBNSV, and other multipartite viruses, actually propagate [10]? More specifically, do they move within host plant and from plant to plant through aphid vectors as nucleoprotein complexes (non encapsidated nucleic acid) or alternatively as assembled viral particles [12]?

Here, we address this question by reporting a structural characterisation of the FBNSV particles at quasi-atomic resolution (3.2 Å) by cryogenic-electron-microscopy (cryo-EM) single-particle analysis. Based on the analysis of the atomic model of the viral particle, the residues involved in CP inter-subunit interactions have been identified. A set of site-directed mutations at one of the CP subunit-subunit interfaces have then been designed in order to prevent the formation of fully assembled particles. These mutations are predicted to hamper the stability of the capsid assembly with negligible effects on the CP fold, and are not directly involved in the observed CP-DNA interaction sites. We first verified, through bacterial expression, that one of the mutated versions of the coat protein could assemble into pentamers but not into virus-like particles (VLPs). Then, we introduced these mutations into the FBNSV genome. The modified coat proteins could be expressed and accumulated into plant tissues; however, the long distance movement of the virus, and thus systemic infection of the host plant, was abolished in all cases.

Based on these results we propose that the viral genome does not propagate within the plant vascular system under the form of uncoated DNA molecules or DNA-CP complexes, but rather moves long distance as assembled viral particles.

## Results

### The cryo-EM structure of FBNSV reveals details of the CP icosahedral assembly and some CP-DNA interaction sites

Suspensions of FBNSV particles were purified from infected faba bean plants. The purified sample contained a mixture of non equivalent virus particles encapsidating different genomic segments. Using quantitative PCR, we first estimated the relative proportion of the various segments in the purified viral population to 4.25, 7.83, 15.58, 3.33, 6.84, 11.56, 20 and 30.61%, for C, M, N, R, S, U1, U2, and U4, respectively. This is consistent with previous reports of unequal frequency of the distinct segments accumulating in infected plants, with N, U1, U2 and U4 being more frequent and C, M, R and S being rarer in *Vicia faba* hosts [13,14]. This heterogeneous particle population, vis-a-vis DNA content, was then investigated by cryo-EM single particle analysis. The images showed a homogeneous spread of spherical particles having a diameter ≈18 nm (S1A Fig). A total of 104,966 particles were extracted from 791 images and processed as described in Materials and Methods. Since the set of imaged particles was taken from a mixture of virions containing different genomic segments, image analysis was initially focused on the capsid shell by masking out the particle inner region containing the DNA segments. These were included later in a second round of image processing.

The initial two-dimensional (2D) classification of binned (2.42 Å/pixel) particle images revealed evident structural heterogeneity (Fig 1). Two groups of 2D class averages were distinguished. The first group (red frames in Fig 1), encompassing 38,728 particles, shows compact full-capsid assemblies. These particles were selected for further processing and 3D reconstruction. The second group of 2D class averages exhibits loose assemblies composed by globular domains which correspond (in the light of the reconstructed structure of the full capsid described below) to pentameric CP assemblies (capsomeres). In this second group, the most populated 2D class (green frame in Fig 1) corresponds to 4,869 particles of a size comparable to that of a single pentameric capsomere, though it is not excluded that these particles could correspond to a contaminant. Some 2D classes show viral particles having lost one or several

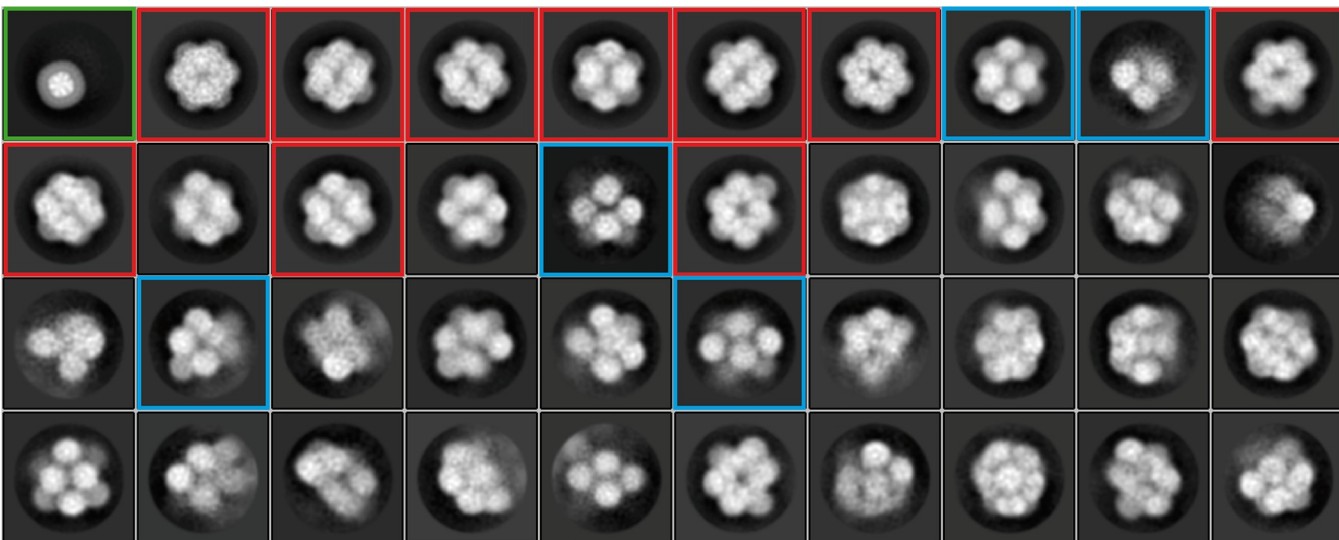

**Fig 1. Two-dimensional class averages computed from the initial set of particles.** 2D class averages are sorted in row-major order according to the number of images contained within each class (top left class: highest number of particles; bottom right class: lowest number of particles). The 2D class average within the green frame represents an isolated pentameric capsomere and encompasses 4,869 particles. The 2D class averages within red frames correspond to compact, full-capsid assemblies that were used for further processing (38,728 particles). The 2D class averages within blue frames are examples of loose viral particles or partial assemblies of pentameric capsomeres.

pentameric capsomeres (blue frames in Fig 1). This indicates that the purified FBNSV particles tend to dissociate—at least in the purification conditions of our samples—by losing pentameric capsomeres, suggesting that CP-CP interactions are stronger at the fivefold interfaces and give rise to more stable assemblies (pentameric capsomeres) than interactions at the two- or three-fold interfaces of the icosahedron. This is consistent with the features of the CP-CP interfaces observed in the atomic structure of the full capsid (see below). At this stage, a 3D classification focused on the capsid shell didn't allow us to distinguish structural differences among viral particles.

Image processing of the best resolved 3D classes enabled us to compute a 3D map of the FBNSV electron scattering density—with imposed icosahedral symmetry—(Electron Microscopy Data Bank entry EMD-10097) from 5,156 non-binned (1.21 Å/pixel) selected particles at an average resolution of 3.2 Å (Figs 2 and S1). According to a local resolution estimation (S1C and S1D Fig) the interior of the capsid layer is resolved up to 2.9 Å, and interpretable details (side chains) of the CP subunits are visible (Fig 2B). A sectional view of the map (S1D Fig) reveals additional densities on the inner face of the capsid shell in close proximity of the five-fold interfaces.

An atomic model of the capsid (Protein Data Bank entry PDB-6S44) was built and refined based on the cryo-EM map, spanning residues 27 to 172 (the C-terminus) for all sixty equivalent CP subunits composing the icosahedral assembly (Figs 2C and 2D and 3 and S2 and Table 1). The highly positively-charged N-terminal stretch (residues 1 to 26) of the CP, which is predicted to be dynamically flexible (see Materials and Methods) and likely buried within the genetic material at the interior of the viral particles, is not visible in the reconstructed density and thus not included in the deposited atomic model. If not otherwise stated, the terms "capsid subunit" and "CP fold" will hereafter make reference to the modelled part—residues 27 to 172—of the CP. The wwPDB official validation report for entries PDB-6S44 and EMD-10097 is available on line (http://dx.doi.org/10.2210/pdb6s44/pdb).

The CP fold (Fig 2D) is a typical wedge-shaped jelly-roll domain—the most prevalent fold among viral CPs [15]—composed by two packed, four-stranded, anti-parallel β-sheets (named

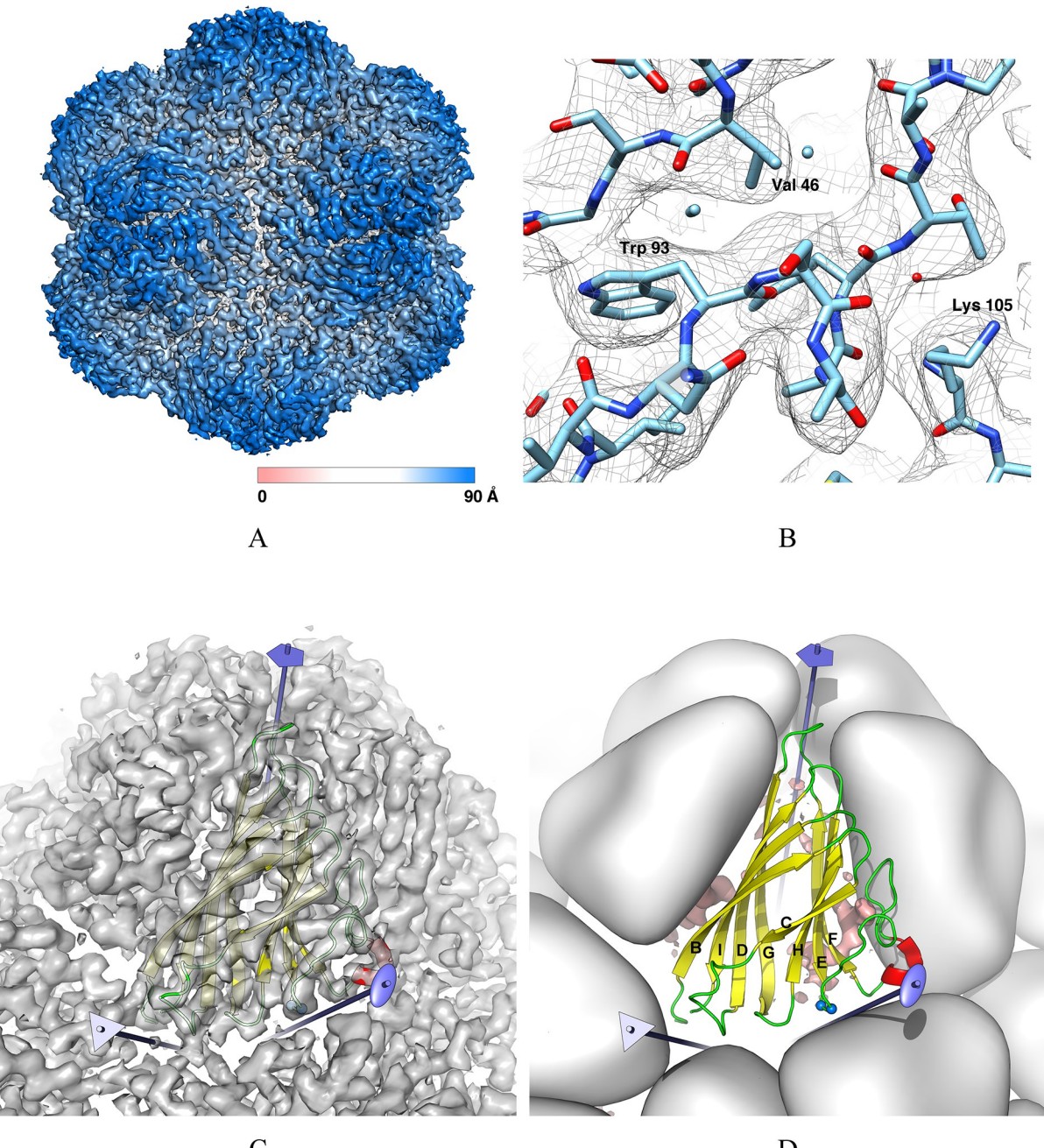

**Fig 2. Cryo-EM 3D reconstruction of FBNSV and atomic model of the FBNSV capsid protein.** (A) An isosurface of the FBNSV scattering density (sharpened map) reconstructed at 3.2 Å resolution (EMD-10097). The surface is coloured according to the distance from the particle centre. (B) A detail of the cryo-EM map and the refined CP atomic model (PDB-6S44). (C) Close-up of the cryo-EM map. The backbone model of one CP subunit is shown as a coloured ribbon. The five-, three- and twofold symmetry axes closest to the highlighted subunit are drawn and coloured in light blue. (D) Ribbon representation of one CP subunit. Secondary structures are coloured in yellow (β-strands) and red (α-helix). The β-strands of the jelly-roll fold are labelled B through I according to classic usage. The surrounding subunits are represented as low-resolution surfaces and coloured white. The five-, three- and twofold symmetry axes closest to the central subunit are drawn and coloured in light blue. The position of Ser 87 and Ser 88, which have been targeted for site directed mutagenesis, is indicated by the two blue spheres at the base of the jelly-roll wedge in proximity of the twofold axis. The DNA residual density observed in the cryo-EM reconstruction is also shown (salmon isosurface).

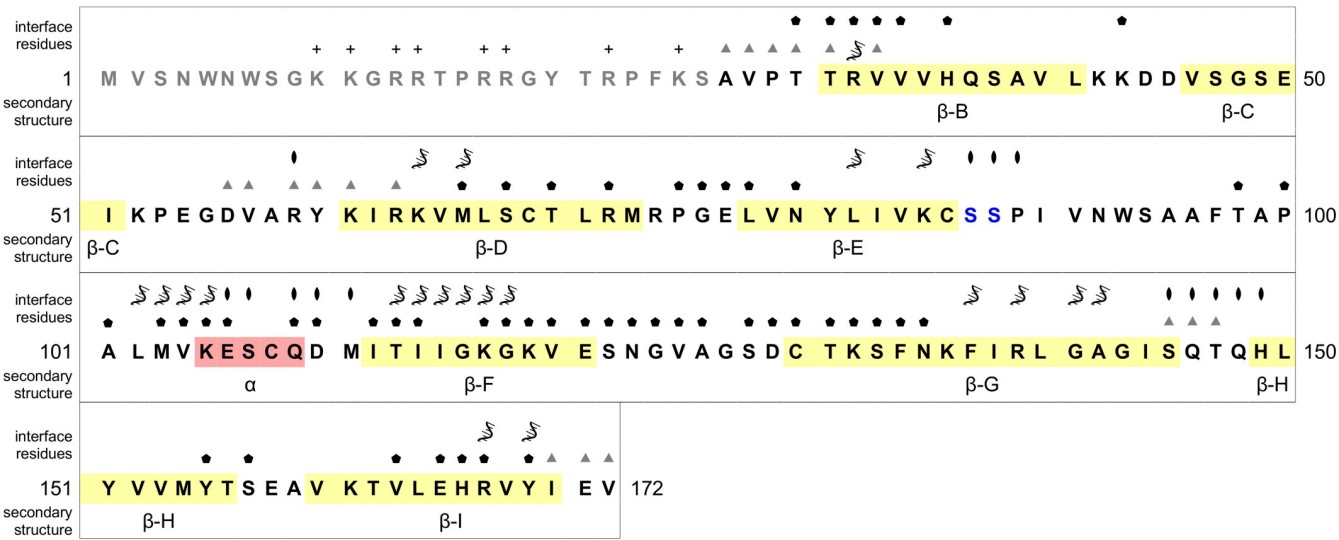

**Fig 3. Full sequence of the FBNSV CP.** The twenty-six N-terminal residues not visible in the 3D cryo-EM map are in grey font. Secondary structures are labelled and coloured in yellow (β-strands) and red (α-helix). The two serine residues targeted for site directed mutagenesis are in blue font. Residues at the CP-CP or CP-DNA interfaces are indicated by markers.

BIDG and CHEF according to classic labelling of jelly-roll β-strands using letters B through I [16]). The loops connecting the β-strands are relatively short (3 to 9 residue long), with the exception of the 25 residue EF connection (residues 87–111) which largely buries the CHEF β-sheet. Each jelly-roll subunit is oriented so that the lateral sides of two β-strands (F and G) face the capsid interior and the closest fivefold axis, while the two strands (B and C) on the opposite side of the jelly-roll contribute to the outer surface of the capsid. This gives rise to an arrangement of jelly-roll domains which is similarly observed in the genus *Albetovirus* [17–20] and (partly) in the family *Geminiviridae* [21], but different from the known structure of other $T = 1$ icosahedral viruses (see the Discussion for a more detailed structural comparison of FBNSV with other viruses).

The CP subunits interact with each other most extensively at the interfaces generated by the icosahedral fivefold symmetry axes (subunit-subunit interface area: 950 Å$^2$) (Table 2 and Figs 2D and 3 and S3), where the BIDG sheet of each subunit is tightly packed against part of the CHEF sheet and the intervening EF connection of an adjacent subunit. Also, residues from all four loops at the top edge of the jelly-roll wedge are involved in fivefold interactions. Fewer interactions are observed at the twofold interfaces (interface area: 393 Å$^2$) (Table 2 and Figs 2D and 3 and S3). These involve loops at the base of the wedge, including the two serines (residues 87 and 88, Fig 4) at the beginning of the EF connection. The fewest contacts between subunits are found at the threefold interfaces (interface area: 301 Å$^2$; Table 2 and Figs 2D and 3 and S3) and involve the first visible residues of the N-terminal stretch, which appear to emerge from the capsid interior, the adjacent C-terminal stretch, which points towards the capsid interior, and two loops at the base of the jelly-roll wedge. Theoretical calculations (based on PISA [22,23]) suggest that these interfaces are likely sufficient to stabilise single pentameric but not trimeric nor dimeric assemblies of CP jelly-rolls in solution. This is consistent with the observed propensity (within our samples of purified FBNSV) of pentameric capsomeres to detach from the capsid assembly (Fig 1).

The additional density visible in close proximity to the inner face of the capsid has been ascribed to the residual average signal from different genome segments. Though less resolved

**Table 1. Cryo-EM data collection, refinement, and validation statistics.**

| PDB-6S44 and EMD-10097 | |
|---|---|
| **Data collection and processing** | |
| Magnification | 20,000 |
| Voltage | 300 kV |
| Electron exposure | 40 electrons/Å$^2$ |
| Defocus range | −1.5 to −3.0 μm |
| Pixel size | 1.21 Å |
| Symmetry imposed | *I* |
| Particle images (no.) | 5156 |
| Map resolution (FSC 0.143 cut-off) | 3.2 Å |
| Map sharpening *B* factor | -40.08 Å$^2$ |
| **Atomic model refinement** | |
| Resolution range used for refinement | 430.76–3.19 Å |
| Model composition (one icosahedral asymmetric unit) | |
| Non hydrogen atoms | 1,120 |
| Protein residues | 146 |
| *B* factors min \| mean \| max | 69 Å$^2$ \| 83 Å$^2$ \| 106 Å$^2$ |
| R.m.s. deviations and Z-scores | |
| Bond lengths | 0.01 Å (Z = 0.405) |
| Bond angles | 1.23˚ (Z = 0.554) |
| MolProbity validation | |
| MolProbity score (full icosahedral capsid) | 2.86 |
| Clashscore (full icosahedral capsid) | 10 |
| Poor rotamers | 11% |
| *cis* peptides | 0% |
| twisted peptides | 0% |
| Ramachandran plot | |
| Favoured | 90.3% |
| Allowed | 9.7% |
| Disallowed | 0.0% |

(>4.5 Å) than the density of the surrounding protein layer, the residual genome density is comparably strong in the neighbourhoods of the CP-CP fivefold interfaces, where a putative aromatic base—sandwiched between the BIDG and CHEF sheets of neighbouring CP subunits—protrudes towards the main-chain atoms of residues 102 to 104 of the EF connection (Fig 5). A total of twenty-one residues per CP subunit has been identified as being in contact with the residual genome density (Fig 3).

Having established a reliable reference map of the FBNSV particles at 3.2 Å resolution, a second round of 3D classifications of the selected (38,728) viral particles was carried out. The particle images were re-extracted at high resolution (1.21 Å/pixel) with the aim of capturing and distinguishing further structural details. Despite several attempts—either by masking out or including the particle inner region containing the DNA segments (S1B Fig)—we were not able to detect resolvable heterogeneities within the set of selected particles. In addition, localised asymmetric classification / reconstruction [25] attempts, focussed around single pentameric capsomeres or single CP-CP five-fold interfaces, were not successful. We conclude that, as they stand, the selected images display viral particles which appear indistinguishable with respect to their DNA-segment content (this point is further discussed in Conclusions).

**Table 2. Analysis of the FBNSV CP-CP interfaces.**

|  | 5-fold interfaces | | 3-fold interfaces | | 2-fold interfaces | |
|---|---|---|---|---|---|---|
|  |  | (probability) |  | (probability) |  | (probability) |
| CP-CP interface area / Å$^2$ | 950 | (70.2%) | 301 | (10.4%) | 393 | (14.7%) |
| Solvation energy / kcal mol$^{-1}$ | -6.4 | (53.3%) | -2.6 | (20.2%) | -3.1 | (21.7%) |
| Total binding energy / kcal mol$^{-1}$ | -11.3 | (75.3%) | -3.1 | (10.2%) | -4.0 | (13.5%) |
| Hydrophobic P-value | 0.4 | (25.8%) | 0.3 | (27.7%) | 0.4 | (19.5%) |
| Number of hydrogen bonds | 7 | (41.8%) | 1 | (10.5%) | 2 | (15.1%) |
| Number of salt bridges | 5 | (39.7%) | 0 | (14.9%) | 0 | (14.9%) |

Residue contacts and theoretical values of thermodynamic parameters calculated using PISA [22,23]. Values between parentheses indicate the probability [24] for an interface with given or higher (lower for the P-value) value of the respective parameter to be part of an assembly of folded macromolecular chains in aqueous solution.

## Introduction of charge repulsion at the capsid twofold interfaces by site-directed mutagenesis suppresses FBNSV systemic infection of plants

As stated in the introduction, an important question concerning the biology of FBNSV is whether the genome segments propagate within host plant as nucleoprotein complexes (non encapsidated nucleic acid) or assembled virus particles. We thus assessed the infectivity of

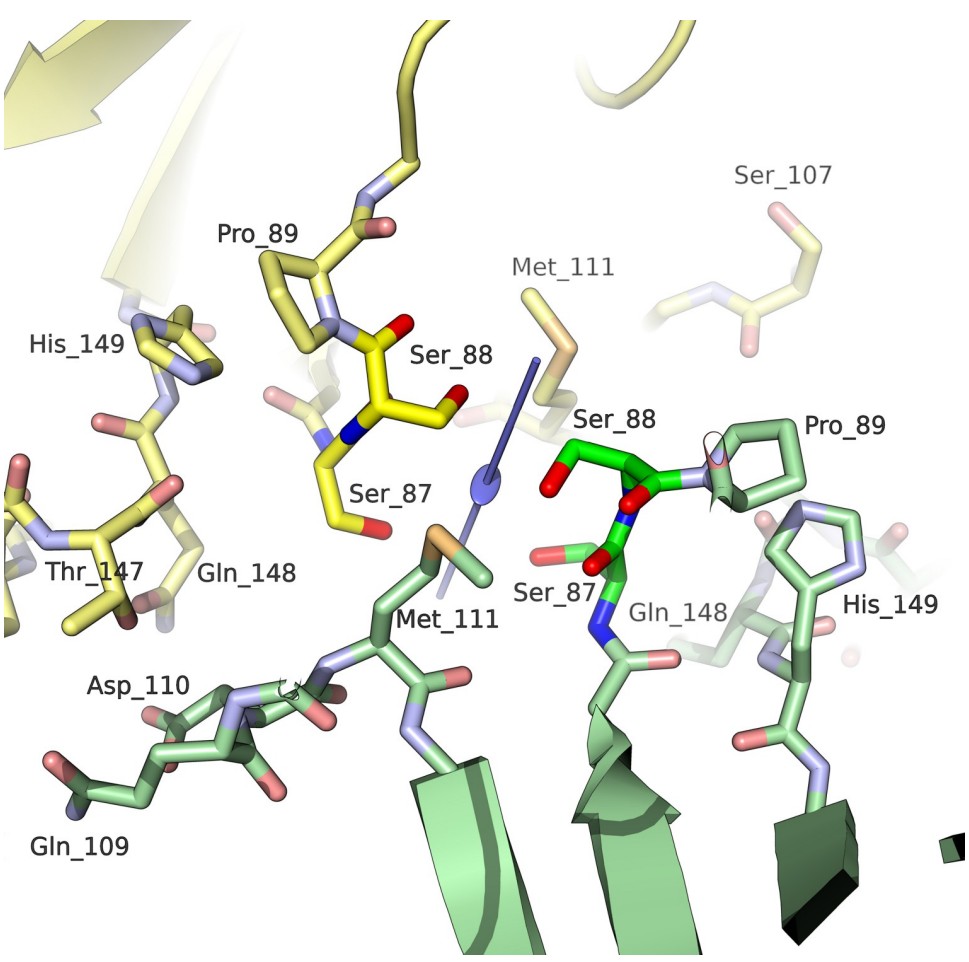

**Fig 4. Close-up view of the twofold interface between CPs including Ser 87 and Ser 88.** The twofold symmetry axis is shown in light blue colour at the centre of the picture.

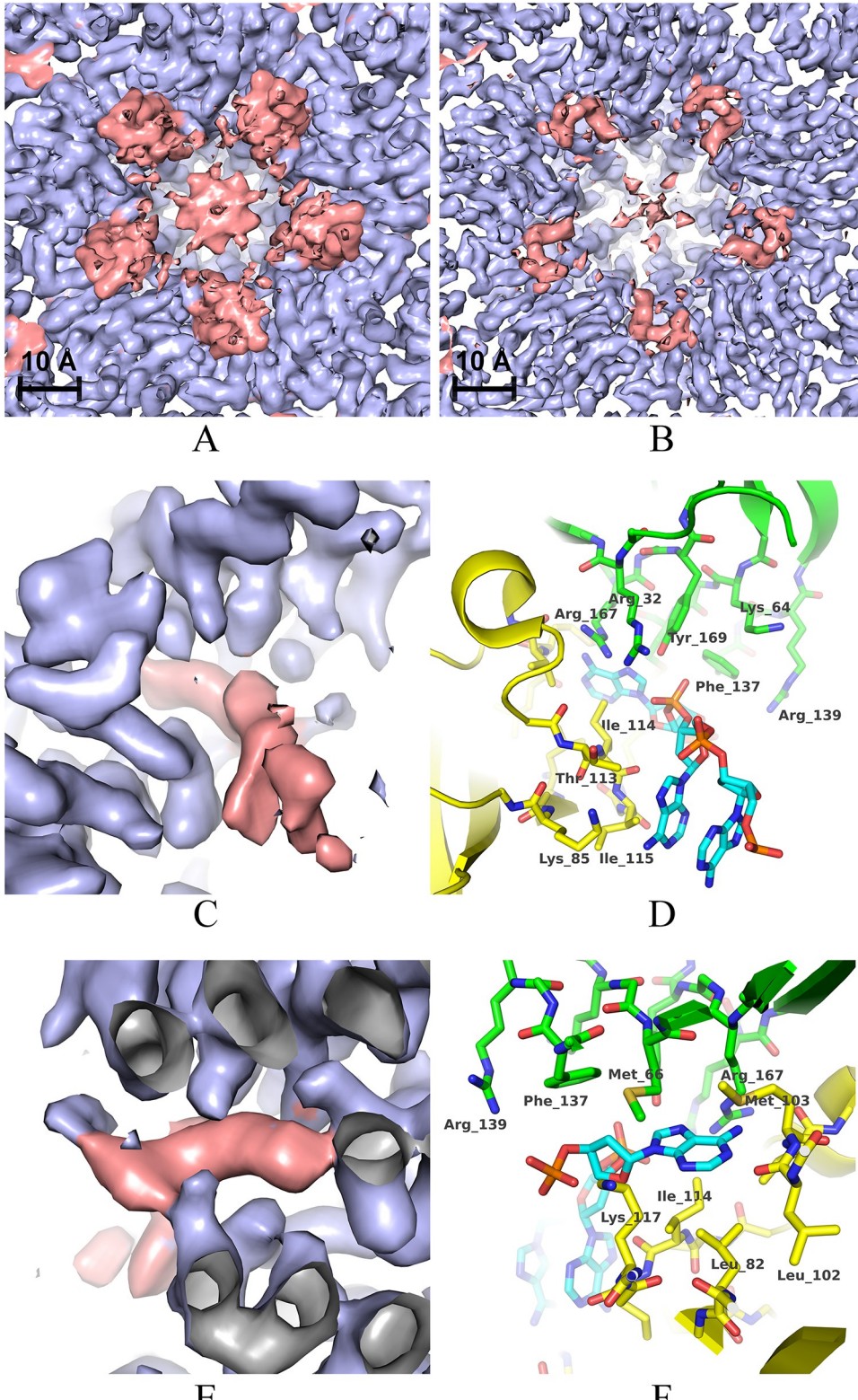

**Fig 5. Residual genomic density in proximity of the fivefold CP-CP interfaces.** (A) and (B): unsharpened and sharpened map respectively, viewed from the capsid interior along a fivefold symmetry axis. Density ascribed to the protein shell is coloured in light blue and the residual density ascribed to DNA is coloured in salmon. (C) and (E): two views of a detail of the DNA density showing a putative aromatic base sticking into a fivefold CP-CP interface. (D) and

(F): atomic models corresponding to the densities depicted in (C) and (E). The refined atomic models of two adjacent CP subunits at the fivefold interface are represented with yellow and green carbon atoms. A tentative DNA fragment model (molecule with cyan carbon atoms) has been included for illustration purpose.

mutants that we predicted to fail to assemble virus particles. We targeted two residues involved in the twofold interfaces of the virus particle, serines 87 and 88 (Fig 4), and replaced either or both of them by charged residues (aspartate, glutamate or arginine). These substitutions introduce electrical repulsion between adjacent subunits with otherwise predicted minute effects on the CP folding and on the stability of pentameric capsomeres. Notably, the mutated residues are not in contact with the observed residual genomic density at the fivefold interfaces.

Seven single or double mutants were generated in the S-segment and faba-bean plant sets were agroinoculated with one of these mutants plus the other seven segments. Table 3 summarises the infection results obtained for each of the mutants. None of them could ever systemically infect faba beans, despite the high number of inoculated plants per experiment and the number of experimental repeats (two to four) per mutant. Occasionally, a few plants inoculated with some of the mutants showed symptoms of systemic infection. In all cases, however, the CP coding sequence amplified from these plants corresponded to the wild type sequence, indicating that systemic infection is possible solely when the mutation spontaneously reverts to wild type (which presumably occurs at the site of agroinoculation).

**Table 3. Infectivity of CP mutants of FBNSV.**

| Experiment number[a] | Mutant ID[b] | Number of systemic infections[c] | Positive control[d] | Reversion to wild type[e] |
|---|---|---|---|---|
| 1 | D87-S88 | 1/48 | 8/45 | Yes |
| 2 | D87-S88 | 1/40 | 22/48 | Yes |
| 3 | E87-S88 | 0/48 | 12/48 | N/A |
| 4 | E87-S88 | 0/48 | 6/48 | N/A |
| 5 | R87-S88 | 5/48 | 8/45 | Yes [f] |
| 6 | R87-S88 | 1/48 | 11/47 | Yes |
| 7 | S87-D88 | 0/42 | 22/48 | N/A |
| 8 | S87-D88 | 1/44 | 11/47 | Yes |
| 9 | S87-E88 | 0/43 | 12/48 | N/A |
| 10 | S87-E88 | 0/46 | 21/90 | N/A |
| 11 | S87-R88 | 0/47 | 13/86 | N/A |
| 12 | S87-R88 | 0/45 | 7/48 | N/A |
| 13 | D87-D88 | 0/48 | 13/86 | N/A |
| 14 | D87-D88 | 0/45 | 7/48 | N/A |
| 15 | D87-D88 | 0/47 | 7/24 | N/A |
| 16 | D87-D88 | 0/90 | 7/96 | N/A |

[a] One test of one mutant is here considered as one experiment.

[b] See Materials and Methods.

[c] Number of systemically infected plants over the total number of inoculated plants.

[d] A positive control means inoculation with all 8 genomic segments of the original pBin19 infectious clone [26]. The number of systemically infected plants over the total number of inoculated plants is reported in this column. Note that repeated numbers in this column correspond to the same positive control experiment carried out when more than one mutant were tested on the same date.

[e] Each plant systemically infected after inoculation with one of the mutants was submitted to PCR amplification of the FBNSV CP coding sequence and Sanger sequencing.

[f] Note: for this mutant, four out of the five systemically infected plants contained the revertant only (no detectable mutant sequence), and one out of five contained a mixture of mutant and wild type.

**Table 4. Infectivity of a forced-reversion of a FBNSV CP mutant.**

| Experiment number[a] | Mutant ID[b] | Number of systemic infections[c] | Positive control[d] | Negative control[e] | Reversion to wild type[f] |
|---|---|---|---|---|---|
| 1 | E/S87-Rev | 5/48 | 7/45 | Not tested | N/A |
| 2 | E/S87-Rev | 6/48 | Not tested | 0/48 | N/A |
| 3 | E/S87-Rev | 9/22 | 11/22 | 1/23 | Yes |
| 4 | E/S87-Rev | 9/63 | 11/68 | Not tested | N/A |

[a, b, c, d] Same as in Table 3

[e] The negative control corresponds to inoculation of plants with the non reverted mutant E87-S88.

[f] Each plant of the negative control that was systemically infected was submitted to PCR amplification of the FBNSV CP coding sequence and Sanger sequencing.

To make sure that the lack of infectivity of the mutants was a direct consequence of the mutations introduced in the CP coding sequence and not due to off-target mutations that might have spontaneously occurred anywhere else in the cloned viral segment, we further mutagenized the mutant E87-S88 (see Materials and Methods for an explanation of the mutant names) back to the wild type sequence (E/S87-rev in Table 4). This forced reversion restored the infectivity of the clone at a rate comparable to that of the positive control.

We further verified, on one mutant example, that the modified segment S could replicate and that the corresponding modified CP was produced and could accumulate in plants tissues, as described in Materials and Methods. S4 Fig shows unaffected replication of the mutated segment S87-R88 and accumulation of the corresponding mutated CP into agroinfiltrated leaves, though seemingly in lower amounts when compared to the wild type CP control. We conclude that the lack of infectivity of the CP-mutants cannot be attributed to a default in replication nor to the complete absence of the mutated CP.

Finally, to validate the prediction that replacement of Ser 87 and/or Ser 88 by charged residues hinders the formation of full virus particles with minimal effect on the CP fold and on the stability of pentameric capsomeres, we overexpressed (in *Escherichia coli*) and purified two recombinant versions of the CP, and analysed putative self-assembly by negative-stain electron microscopy (Fig 6). The first version (ΔN-CP) corresponds to the wild-type CP lacking the flexible N-terminal stretch (residues 1 to 26). The second version (ΔN-D87-D88-CP) is a derivative of ΔN-CP where serines 87 and 88 are replaced by aspartates. In these two versions, the N-terminal stretch was deleted because its highly positive charge is likely to impede the formation of full icosahedral capsids when the viral ssDNA segments are absent [27]. It should be noted that we did try to overexpress the full-length CP—both wild-type and double mutation—but final protein yield was too low. Negative stain images of ΔN-CP (Fig 6A) revealed that the protein forms oligomeric aggregates of different sizes up to spherical VLPs of about 20–25 nm in diameter. On the contrary, no VLPs were observed in samples of purified ΔN-D87-D88-CP, for which the size and shape of the largest observed assemblies (round particles indicated by white arrows in Fig 6B) are consistent with those of pentameric capsomeres. These observations confirm our hypothesis that replacing serines 87 and 88 by charged residues alters the interactions between pentameric capsomeres and impairs higher order assembly and the formation of complete capsids.

## Discussion

### Nanoviruses are structurally closely related to geminiviruses

A number of atomic structures have been reported for viruses that organise their capsid using single-domain jelly-roll subunits arranged in $T = 1$ icosahedral assemblies. These include

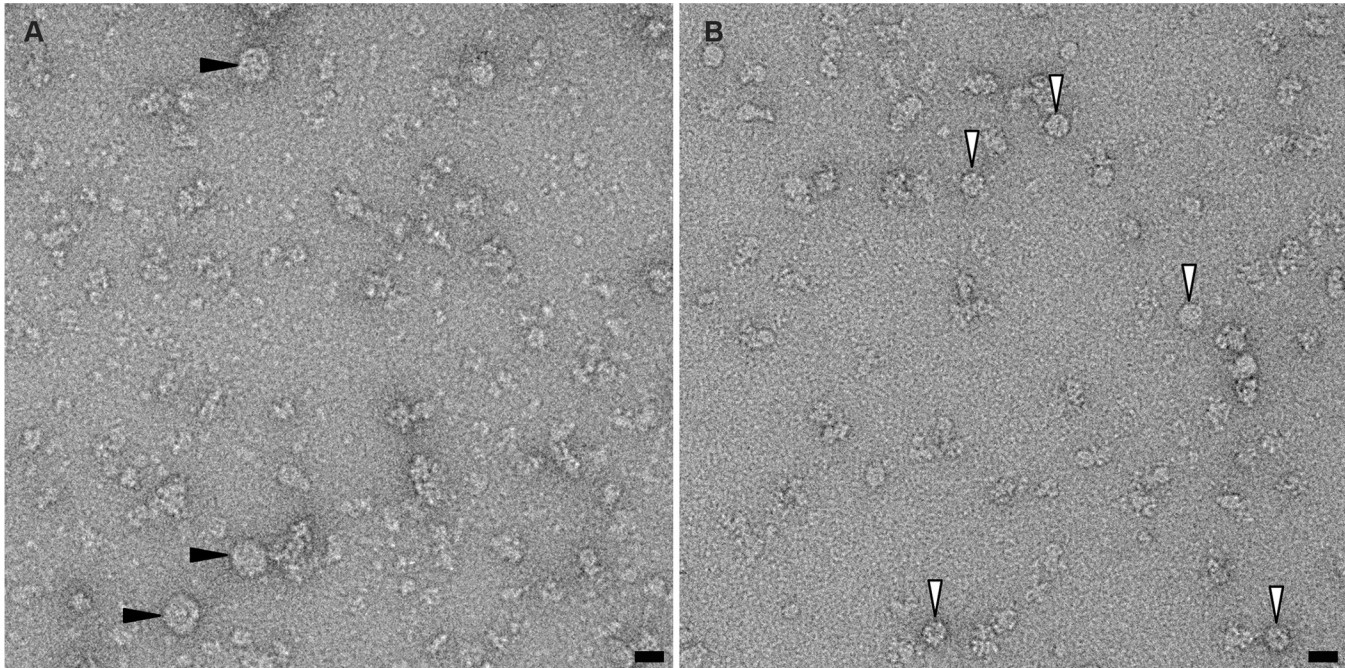

**Fig 6. Negative-stain electron-microscopy images of purified recombinant CP overexpressed in *E. coli*.** (A) ΔN-CP construct (residues 27–172). VLPs are visible in the sample (some of them are indicated by black arrows). (B) Double mutant ΔN-D87-D88-CP. No VLPs were observed. Smaller assemblies, likely pentameric capsomeres, are visible (white arrows). Scale bars in (A) and (B): 20 nm.

members of the families *Circoviridae* [28,29] (animal viruses with monopartite, circular, ssDNA genome), *Sarthroviridae* [30] (animal viruses with monopartite, linear, ssRNA(+) genome), and species *Tobacco albetovirus 1* (satellite tobacco necrosis virus, STNV) [17–20], *Panicum papanivirus 1* (satellite panicum mosaic virus, SPMV) [31,32] and *Tobacco virtovirus 1* (satellite tobacco mosaic virus, STMV) [33,34] (plant viruses with monopartite, linear, ssRNA(+) genome). It has long been recognised that—despite the globally conserved, simple jelly-roll fold of their CP—these small viruses organise their icosahedral shells in prominently different manners [35] (a comparison between the structure of these viruses and FBNSV is illustrated in Fig 7). In addition, member species of the family *Geminiviridae*—on the example of ageratum yellow vein virus (AYVV) [21]—possess ssDNA(+) circular genome packaged into typical, geminate particles composed of two incomplete $T$ = 1 icosahedra (Fig 7). The solved structures of more distantly related $T$ = 1 icosahedral viruses or subviral particles—including members of *Adenoviridae* [36,37], *Microviridae* [38–42] and *Parvoviridae* [43], which also contain the jelly-roll fold—reveal a more complex (as compared to FBNSV), multi-domain or hetero-oligomeric organisation of the capsid shell.

A comparison of FBNSV with those structures, and the recently reported [27] structural analysis of the homology-based model of banana bunchy top virus indicate that nanoviruses are structurally most closely related to geminiviruses (Fig 7), despite the geminate nature of the geminivirus capsid. FBNSV exhibits a CP packing which is similar on the whole to that of AYVV (half capsid), with extensive contacts at the (pseudo) fivefold interfaces. A comparison of single pentameric capsomeres from FBNSV and AYVV (Fig 7D) shows that the relative orientations/positions of their subunits differ by less than 3.0˚ / 2.7 Å. However, the distances of the pentameric sub-assemblies from the icosahedron centre are larger in AYVV by +10 Å, accommodating the loops at the base of the jelly-roll wedge (including the long EF connection) which are also longer.

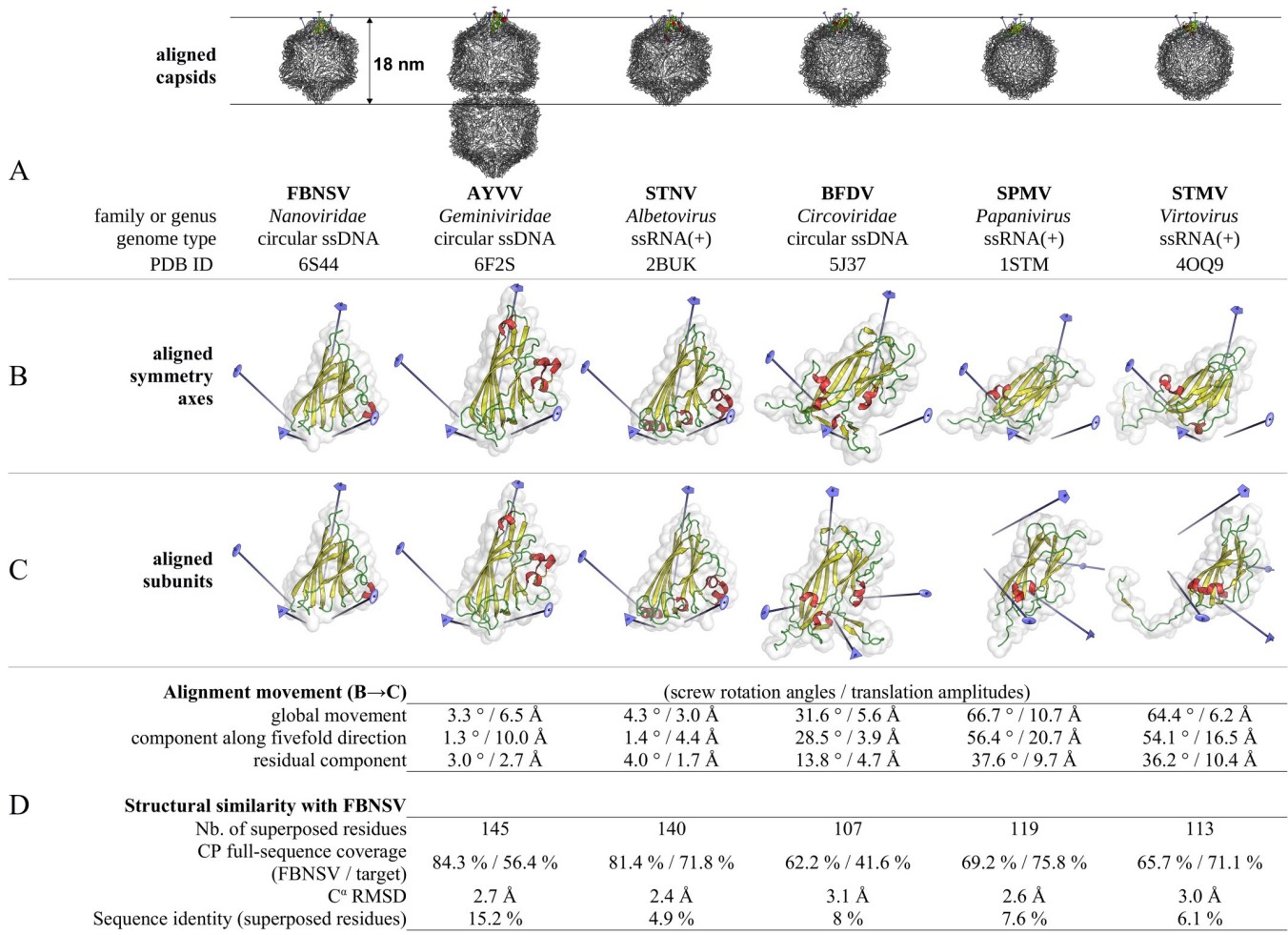

| | | | | | |
|---|---|---|---|---|---|
| family or genus | **FBNSV** *Nanoviridae* | **AYVV** *Geminiviridae* | **STNV** *Albetovirus* | **BFDV** *Circoviridae* | **SPMV** *Papanivirus* | **STMV** *Virtovirus* |

Fig 7. Comparison of the FBNSV capsid structure with those of other T = 1 icosahedral (or geminate pseudo-icosahedral in the case of AYVV) viruses.

**Alignment movement (B→C)** (screw rotation angles / translation amplitudes)

| | | | | | |
|---|---|---|---|---|---|
| global movement | 3.3 ° / 6.5 Å | 4.3 ° / 3.0 Å | 31.6 ° / 5.6 Å | 66.7 ° / 10.7 Å | 64.4 ° / 6.2 Å |
| component along fivefold direction | 1.3 ° / 10.0 Å | 1.4 ° / 4.4 Å | 28.5 ° / 3.9 Å | 56.4 ° / 20.7 Å | 54.1 ° / 16.5 Å |
| residual component | 3.0 ° / 2.7 Å | 4.0 ° / 1.7 Å | 13.8 ° / 4.7 Å | 37.6 ° / 9.7 Å | 36.2 ° / 10.4 Å |

**D** Structural similarity with FBNSV

| | | | | | |
|---|---|---|---|---|---|
| Nb. of superposed residues | 145 | 140 | 107 | 119 | 113 |
| CP full-sequence coverage (FBNSV / target) | 84.3 % / 56.4 % | 81.4 % / 71.8 % | 62.2 % / 41.6 % | 69.2 % / 75.8 % | 65.7 % / 71.1 % |
| Cα RMSD | 2.7 Å | 2.4 Å | 3.1 Å | 2.6 Å | 3.0 Å |
| Sequence identity (superposed residues) | 15.2 % | 4.9 % | 8 % | 7.6 % | 6.1 % |

**Fig 7. Comparison of the FBNSV capsid structure with those of other $T$ = 1 icosahedral (or geminate pseudo-icosahedral in the case of AYVV) viruses.** All virus acronyms are explained in the text, except BFDV: beak and feather disease virus. (A) Outline of the whole capsid structures. The capsids are aligned to each other with respect to their symmetry axes (half-capsid pseudo-symmetry axes in the case of AYVV). A single CP subunit in each structure is highlighted in colours. The five-, three- and twofold axes closest to the highlighted subunits as well as the second closest twofold axis are drawn. The vertical direction corresponds to a fivefold direction. (B) Side-by-side representation of single CP subunits oriented so that the capsid (pseudo)symmetry axes are aligned to those of FBNSV (which is oriented as in Fig 2D). The five-, three- and twofold axes closest to each subunit as well as the second closest twofold axis are drawn. Note that the twofold axis closest to the FBNSV, AYVV and STNV subunits (on the bottom right of each subunit picture) does not correspond to the twofold axis closest to the BFDV, SPMV and STMV subunits (on the left of each subunit picture). (C) Side-by-side representation of single CP subunits oriented after structural superposition onto the FBNSV CP subunit (FBNSV oriented as in Fig 2D). (D) Details of the virtual superposition movement (B→C) decomposed into two movements (one around the fivefold direction and the residual around a perpendicular direction) and statistics of the structural comparison with FBNSV. The arrangement of the CP subunits relative to each other in FBNSV resembles that of STNV and AYVV (half capsid). On the other hand, and despite the overall conserved jelly-roll fold, a significantly different arrangement is observed in BFDV, SPMV and STMV.

A structural feature observed in FBNSV, partially shared with AYVV, is the lack of any ordered polypeptide stretch upstream the CP jelly-roll domain, in proximity of the threefold axes. The positively-charged N-terminal stretch (residues 1–26) is unresolved in the FBNSV cryo-EM map, and predicted to be dynamically flexible. The number of inter-subunit contacts at the capsid threefold interfaces is consequently quite low. The situation is similar in AYVV, except for the CP subunits at the interface between the incomplete icosahedra, where a number of ordered residues (either eight or twenty-three out of sixty-two composing the positively charged N-terminal stretch) provide for alternative CP-CP interactions at the equator of the geminate capsid [21].

The residual DNA density observed in the FBNSV cryo-EM map highlights some interesting features of the genome organisation and points to further similarities between nanoviruses and geminiviruses. Despite being the average signal from sixty different sites per particle and coming from a set of virions containing different genomic segments, the residual genome density is significantly strong, comparable to—though less resolved than—that of the surrounding polypeptide layer. This indicates the presence of a highly repeated number of DNA-CP interaction sites which are seemingly highly conserved among virions containing different genomic segments. The DNA-CP interaction sites are located at the fivefold interfaces close to the base of the jelly-roll wedges. Similar zones of interaction with the encapsided DNA were previously observed in AYVV [21], though in that case no unstacked aromatic base was reported to stick into the CP fivefold interfaces (as appears to be the case for FBNSV).

The observed structural, and more generally phylogenetic [44], proximity of nanoviruses to geminiviruses points to the intriguing question of whether nanoviruses would be able to adopt alternative encapsided forms of their multipartite genome, in particular by forming geminate capsids capable of hosting more than one DNA segment (the size of a nanovirus segment, $\approx$1 kb, compared to the size of a geminivirus segment, $\approx$2.5–3.0 kb, indicates that there would be room for two to three nanovirus segments). CP sequence differences between nanovirus and geminivirus do not support this hypothesis. First, the CP N-terminal extensions (about 45 to 70 residues in all known geminiviruses), with residues 41–63 of AYVV adopting alternative conformations and leading to the formation of the AYVV geminate interface, are systematically shorter in nanoviruses (26 residues in FBNSV and even less in other nanoviruses). Second, the GH loop, which participates to the formation of the AYVV geminate interface through the reported hydrophilic patch His214-Asn215 [21], is much shorter in all known nanoviruses so that an AYVV-like fusion of two incomplete icosahedra seems structurally impossible. Consistently, it should be noted that geminate assemblies of FBNSV have never been observed in our cryo-EM samples.

## Structure-assisted mutagenesis suggests that particle assembly is required for systemic movement of FBNSV within host plants

Our mutagenesis experiments on the FBNSV capsid demonstrate that the substitution (within the S segment of FBNSV DNA) of the sequences coding for CP Ser 87 and/or Ser 88 for those of charged amino-acids abolishes the viral capacity to systemically infect faba-bean host plants. Importantly, we verified that the mutated viral DNA could replicate and that the corresponding mutated protein could accumulate in host plant cells. Thus, the lack of systemic infection cannot be trivially attributed to the absence of segment S and/or coat protein. Due to their position at the base of the jelly-roll wedge, and considering the electron-microscopy images of the recombinant ΔN-D87-D88-CP, which show structures reminiscent of pentameric capsomeres, these mutations have likely negligible effects on the fold of individual CP subunits. Likewise, since the mutated residues are not directly involved in the five-fold interfaces nor adjacent to the observed residual genomic density, these mutations should not affect the observed CP-DNA contacts. In contrast, the mutations have been designed to create a charge repulsion at the two-fold interfaces, sufficient to impair interactions between flanking subunits. The aim of this design was to possibly maintain the formation of pentameric capsomeres, and thus also interaction with DNA-segments, but prevent full particle assembly. Remarkably, both the presence of pentameric capsomeres and the lack of icosahedral assemblies was confirmed in purified recombinant ΔN-D87-D88-CP. We are perfectly aware that unpredicted effects of these mutations on the CP-DNA interactions cannot be totally excluded. Nevertheless, as they stand, our results are consistent with the conclusion that FBNSV

segments require the assembly of virus particles in order to move long distance and infect the host systemically. In favour of this conclusion is also the observation that reversion of some of our CP-mutations could spontaneously occur (particularly those targeting Ser 87). Occasional reversions can be considered as an indication that the mutated virus can indeed replicate within the inoculated cells, but that it can go systemic solely if a reverse mutation of the CP occurs, restoring its genuine structural features.

## Conclusions

When assuming the most probable possibility that FBNSV moves within hosts as assembled virus particles, then an additional interesting question arises: are all viral particles structurally identical, whatever the identity of the encapsidated segment? This question is intriguing because of a sort of controversy (or contrasted way of life) debated in multipartite viruses, that is the way their genetic material is propagated and the integrity of the genome maintained. One possibility is that, somehow, multipartite viruses manage to propagate as one integral genome, meaning at least one copy of each segment, under the form of a large macromolecular complex (sorted propagation). The other possibility is that each segment propagates separately, with no possibility of distinguishing between them (for example if they are individually encapsidated in similar virus particles), and thus that they can only come together at random, with greater chances to maintain the integral genome if the multiplicity of infection is high (random propagation). The respective implications of sorted or random propagation on the understanding of the biology and evolution of multipartite viruses have been discussed and reviewed extensively [9,10,12]. For FBNSV, we provide here two arguments in favour of random propagation as depicted above. First, when averaging increasing amounts of virus particle images, ignoring the nature of their DNA content, we could reach a resolution of the electron scattering density down to 3.2 Å, revealing the absence of major differences (at the protein capsid level) between particles containing distinct segments. Second, in the following round of image processing distinguishing inner density of the viral particles, thus potentially distinguishing between segments, we could not detect resolvable heterogeneities within the set of selected particles. We are aware that the number of images analysed in this second round may be judged too low, and that higher numbers will be required in order to definitely conclude on the absence of minute structural differences related to the identity of the packaged segment. Nevertheless, at this stage, our results suggest that the FBNSV genome propagates within hosts under the form of a population of viral particles containing distinct segments, and that these distinct segments do not induce major differences at the capsid level. Such a situation, where all particles of the viral population are alike, does not support the existence of a large macromolecular complex which would form by specifically sorting and assembling particles each containing a distinct segment. Together with other evidence from totally unrelated approaches [11,45], this further supports the independent propagation of the distinct segments of the FBNSV genome; i.e. the random propagation described above.

## Materials and methods

### Viral infectious clones, plant agroinoculation and agroinfiltration

FBNSV has first been reported in Ethiopia in 1997 from cultivated faba beans, and then maintained in the same host plant species in laboratory conditions by successive aphid-transmission [46]. The corresponding isolate, named FBNSV-ET:Hol:97, was further characterized and used to prepare infectious clones in 2009 [26]. Each of the eight genome segments of this isolate has been inserted as a head-to-tail dimer into the binary plasmid pBin19 to create eight plasmids, together constituting the FBNSV infectious clone [26].

In this study, we used *Vicia faba* of the cultivar "Sevilla" (Vilmorin) for agroinoculation experiments. Ten-day-old plantlets were agroinoculated into the stem with the FBNSV pBin19 infectious clone as earlier described [26] and maintained in growth chambers under a 13/11 hours day/night photoperiod at a temperature of 26/20˚C day/night and 70% hygrometry.

In order to generate a high number of infected plants for further virus purification, symptomatic faba bean plants were checked for the presence of all eight segments by qPCR twenty-one days after agroinoculation. These plants were then used as source plants for aphid transmission. A cohort of approx. 40 individual aphids of the species *Acyrthosiphon pisum* was placed on each source plant for an acquisition period of three days and then transferred as groups of two on receiver *Vicia faba* plantlets of the cultivar "Shambat-75" for an inoculation period of three additional days, prior to insecticide treatment.

Like all nanovirus species, FBNSV is phloem-restricted [1], and agroinoculation into the stem engenders systemic infection. When the same agrobacterium cultures, containing the eight cloned segments, are agroinfiltrated into leaves, the virus cannot efficiently reach deep phloem tissues and the systemic infection aborts. However, it is possible that the infiltrated bacteria can transfer viral DNAs to leaf mesophyll cells, where viral proteins may be produced and local replication may occur. We made use of this possibility to agroinfiltrate the FBNSV infectious clone, in order to detect the local production of the coat protein, including the mutated coat protein derivative S87-R88 described in the next section. Young faba bean cv. Sevilla plants were agroinfiltrated within leaves at the three-leaf stage. Four to six days later, the agroinfiltrated area of the leaves were harvested and ground directly in Laemmli 1x buffer prior to classical protein separation on 15% SDS-PAGE gels [47] and western-blot onto nitro-cellulose membranes. The presence of the coat protein in agroinfiltrated leaf extracts was further immuno-revealed according to classical ECL (electrochemiluminescence) using a primary antibody directed against FBNSV coat protein and a secondary antibody coupled to the alkaline phosphatase. The primary antibody was a monoclonal antibody from mouse as in [6], and the secondary antibody was an anti-mouse IgG-Alkaline phosphatase conjugate from SIGMA.

## Mutagenesis of FBNSV CP

To generate mutated CP, we used the plasmid pCambia 2300-S-SL [48–50] containing one copy of the S segment where the conserved sequence corresponding to the origin of replication is redundant at both extremities. This plasmid has a single copy of the coding sequence of the CP and can thus be easily mutagenized using commercially available kits. Mutations were introduced in the coding sequence of the CP within pCambia 2300-S-SL plasmid replacing the serines at position 87 and/or 88 by charged residues, either glutamate, aspartate or arginine. We herein note the wild type sequence as S87-S88, and the generated single or double mutants are accordingly noted D87-S88, E87-S88, R87-S88, S87-D88, S87-E88, S87-R88 and D87-D88. We also reverted the mutant E87-S88 back to the wild type sequence and this revertant is noted E/S87-Rev. All mutagenesis were performed using the Quick Change II site-directed mutagenesis kit (Agilent), according to the manufacturer instructions, and the sequence of all primers used is available upon request. For testing the effect of segment S mutation on FBNSV infectivity, young plants were agroinoculated as described above by simply replacing the pBin19-S clone with one of the wild type or mutated pCambia 2300-S-SL plasmid.

## Purification of virus particles

One hundred grams of infected Shambat-75 faba bean leaves (solely the three uppermost leaf-levels) were collected and ground with a pestle and mortar chilled in liquid nitrogen. The

resulting powder was then poured into 250 mL of grinding buffer reaching a total volume of approximately 350 mL that was adjusted to the following final composition: 70 mM sodium citrate pH 6, 0.2% ascorbic acid, 0.2% sodium sulphite, 0.3% β-mercaptoethanol, 1.5% cellulase Omozuka-R10 (Duchefa) and 0.4% macerozyme-R10 (Duchefa), and gently shaken overnight at room temperature.

One hundred and twenty-five mL of a 1:1 chloroform/butanol mixture were then added to the plant homogenate, vigorously shaken and centrifuged at 6000 $g$ for 20 minutes at 20˚C. Three hundred mL of the supernatant phase were collected, supplemented with 2% NaCl, slowly mixed at room temperature with 50 mL of 56% (W/V) PEG-6000 solution (8% PEG final), gently agitated for 2 hours at 4˚C, and centrifuged at 18,000 $g$ for 15 minutes at 6˚C. The pellet(s) was resuspended overnight at 4˚C, in a total volume of 55 mL of 20 mM phosphate buffer pH 7.5 containing 1% triton-X-100, using a magnetic stirrer for gentle agitation.

The resuspended pellet(s) was then clarified for 15 minutes at 11,000 $g$ and 4˚C, and the supernatant loaded onto a 20% sucrose cushion in PBS buffer and further centrifuged for 3 hours at 142,000 $g$ and 4˚C. After discarding the supernatant, the pellet(s) was collected in 5 mL total volume of 20 mM phosphate buffer pH 7.5, disrupted with a Potter homogenizer, gently stirred for 1 hour at 4˚C and clarified for 15 minutes at 7500 $g$ and 6˚C. The clarified supernatant was centrifuged at 200,000 $g$ for 3 hours at 6˚C and the pelleted virus particles covered with 0.5 mL of 20 mM phosphate buffer pH 7.5 and let for swelling overnight at 4˚C.

The next morning, the pelleted virus particles were resuspended with a Potter homogenizer, which was rinsed with additional 0.5 mL to recover as much of the virus particles as possible. The resulting volume (1 mL) was loaded onto a 10–40% sucrose gradient prepared in 20 mM phosphate buffer pH 7.5 and centrifuged for 4 hours at 26,000 rpm in a SW41 rotor at 10˚C. The gradient was collected as 1 mL fractions where the presence and quantity of virus particles were checked by negative staining and transmission electron microscopy. The majority of virus particles were generally found in the upper part of the gradient, from fraction 8 to 11, which were each diluted to a final volume of 8 mL with 10 mM phosphate buffer pH 7.5 and centrifuged for an ultimate concentration step at 165,000 $g$ for 10 hours at 12˚C. The purified virus particles were finally resuspended in 150 μL of 10 mM phosphate buffer pH 7.5 and transferred into 0.5 mL low-binding tube for storage at 4˚C.

## Viral DNA extraction and analysis

The extraction of DNA from FBNSV-infected plants and qPCR analysis to check for the presence of various genomic segments were performed as previously described [13]. An aliquot of the suspension of purified virus particles was also extracted to quantify the relative proportion of each of the eight segments in the population used for structural analysis. For all plants showing symptoms after inoculation of any of the mutated S segments, the corresponding PCR-amplified sequence and the presence/absence of the mutation within the CP was verified by Sanger sequencing (Genewiz, France).

Total DNA was similarly extracted from agro-infiltrated leaves and a 15 μL aliquot was treated with DpnI in 25 μL total to eliminate the agro-infiltrated plasmid. 1 μL of this solution was then used as template for rolling circle amplification (RCA) using the TempliPHi Amplification Kit (GE Healthcare) according to the manufacturer instructions. The RCA products, which can only derive from circular DNA and thus from replicated FBNSV segments, were then digested by AatII. Because all eight FBNSV genome segments contain one single copy of this restriction site, such digestion releases a band of approximately 1000 bp from the high molecular weight RCA amplification products, which is easily visible on an agarose gel. This band can be considered as the signature of the replication of the segments because it is

produced from plant extracts only in the presence of the segment R and thus of the replication-controlling protein M-Rep (S4 Fig). To quantify and compare the copy number of segment S, these RCA products were analysed by qPCR as described [48] above for total DNA extract or purified virus suspension.

## Cryo-EM data collection and processing

The viral particles tended to stick to the carbon film rather than being ice trapped within the holes of the cryo-EM grids. For this reason, holey grids covered by a thin layer of carbon were used in order to increase the local density of viruses. Three microlitres of purified viral particles were applied during 2 min to glow-discharged Quantifoil R 2/2 grids covered with a 2–3 nm ultra-thin carbon layer (Quantifoil Micro tools GmbH, Germany), blotted for 1 s and then flash frozen in liquid ethane using the semi-automated plunge freezing device Vitrobot IV operating at 100% relative humidity and 22˚C. Movies were collected using a Polara F30 transmission electron microscope (FEI, Eindhoven, Netherlands) (IBS, Grenoble, France) operating at 300 kV with defocus values ranging from -1.5 to -3.0 μm. The pixel size estimated at the specimen level was 1.21 Å. A total number of 791 movies was recorded using the Gatan Latitude S single particle acquisition software (Gatan inc.) on a K2 summit direct electron detector operating in counting mode. The total exposure time was 5 s corresponding to a dose of 40 e$^-$/Å$^2$. Forty individual frames were collected with an electron dose of 1.0 e$^-$/Å$^2$ per frame.

The image processing workflow followed in this work is described in S5 Fig. The frames of each movie were computationally corrected for drift and beam-induced movement in RELION-3.1 [51]. The contrast transfer function of each micrograph was determined using Gctf program [52]. 104,966 particles were automatically selected using the Gautomatch program [53] and extracted with a pixel size of 2.42 Å. A first round of reference-free 2D classification allowed us to select 38,728 particles that were used to compute a de novo 3D model. A 3D classification was performed into eight classes. Only one 3D class containing 5,156 particles was suitable for further processing. Particles present in this 3D class were then extracted at 1.21 Å per pixel and subjected to a 3D refinement, yielding a map at 4.35 Å resolution. We then proceeded with per-particle defocus refinement and Bayesian polishing, improving the resolution to 3.2 Å. The resolution was estimated by the so-called gold-standard Fourier shell correlation (FSC) using the 0.143 criterion [54]. The final refined map was post-processed using a computed *B*-factor of -40 Å$^2$. Local resolution was estimated using *Blocres* from *Bsoft* package [55]. Further asymmetric classifications and reconstructions (following general guidelines described in [25] and [56]) were focussed around either single pentameric capsomeres or single fivefold interfaces in order to better resolve the residual DNA density, without success. The reconstructed maps of the FBNSV particles have been deposited with EMD accession code EMD-10097.

## Atomic model building, refinement and validation

The FBNSV capsid atomic structure was built based on an initial model composed of the secondary structure core from a single *Geminivirus* capsid subunit (PDB-6EK5) mutated to polyalanine and rigid-body fitted to the cryo-EM reconstruction. FBNSV CP sequence was assigned based on visual identification of bulky side-chain densities and PSIPRED [57] predictions of secondary structure and disordered regions [58]. Loops were built and real-space refined using Coot [59]. A work set of identical polypeptide chains (subset of the full icosahedral assembly) composed of one central CP subunit plus its 7 neighbouring chains was generated and iteratively real-space refined using PHENIX [60] and manually checked and adjusted using Coot. Standard stereochemical plus non-crystallographic symmetry and manually

curated secondary structure restraints were applied during refinement. Grouped isotropic displacement parameters (*B*-factors, one parameter per residue) were refined during the last stages of process. The last refinement cycle was carried out on the whole icosahedral assembly. Finally, a single CP subunit was extracted and deposited—along with annotation of the symmetry operators required to generate the whole capsid—within the wwPDB. Model quality was assessed throughout using Molprobity [61] and the wwPDB validation server. Inter-subunit contacts and theoretical dissociation energies were calculated using PISA [22]. Structural comparison with other virus capsids were carried out using in-house developed software (for icosahedra alignment and rigid-body transformation analysis) and jFATCAT [62,63] (for single subunit-subunit superposition) as implemented on the RCSB PDB website [64,65]. The CP atomic model has been deposited with PDB accession code PDB-6S44.

## Purification of recombinant ΔN-CP and ΔN-D87-D88-CP

The sequences of truncated CP (ΔN-CP, residues 27–172) and truncated double mutant (ΔN-D87-D88-CP) were designed with NdeI and XhoI forward and reverse sites respectively and were used to perform an In-fusion cloning reaction according to the manufacturer's protocol. The digestion of the recipient plasmid pDB (N-terminal fusion) was carried out with the same restriction digestion. Briefly, ΔN-CP and ΔN-D87-D88-CP coding sequences were cloned in an N-terminally His-tagged (His$^6$) plasmid fused to the maltose binding protein which acted as a carrier (pDB-His-MBP-3C-His). After the In-fusion cloning reaction, the Stellar cell was used for plasmid transformation. The confirmed clones of both ΔN-CP and ΔN-D87-D88-CP were then transformed into rosetta (DE3) cells. A small tip was used to inoculate single colonies from each construct into a 15 mL LB medium with 1 μg/mL kanamycin and the cultures were incubated overnight in a 37˚C shaking incubator. Then, all pre-inoculated culture was inoculated into a 2,000 mL flask containing 700 mL LB medium and incubated at 37˚C in a shaking incubator for additional 4 h. Expression was then induced overnight at 25˚C by addition of 1 mM IPTG. Cells were harvested and centrifuged at 6,000 rpm, the bacteria expressing each protein were pelleted, resuspended and then sonicated with 2 min on/off cycles for 4 min at 50% amplitude in 50 mL lysis buffer (50 mM Tris, pH 8.0, 500 mM NaCl, 1 mM DTT) supplemented with 1 mg/mL both lysozyme and DNAse and incubated for 20 min at room temperature. Subsequently, centrifugation of the lysates was done at 4˚C for 30 min at 18,000 rpm. The supernatant was then subjected to separation using a NGC Medium-Pressure Liquid Chromatography Systems (Biorad) using a Ni-NTA affinity resin. The unbound proteins were washed with 50 mL washing buffer (50 mM Tris, pH 8.0, 500 mM NaCl, 60 mM imidazole, 1 mM DTT) and the protein was subsequently eluted with elution buffer (50 mM Tris, pH 8.0, 500 mM NaCl, 350 mM imidazole, 1 mM DTT). Fractions were analysed by SDS-PAGE. Fractions containing more than 95% homogeneous proteins were collected and combined. The MBP carrier protein was cleaved off by overnight digestion with the 3C enzyme (1 mg/mL) and the fraction was dialysed in buffer: 50 mM Tris, pH 8.0, 100 mM NaCl, 1 mM DTT.

The ΔN-CP protein was loaded on a HiLoad 16/60 Superdex 75 column connected with a HisTrap FF column and was pre-equilibrated with a solution of 50 mM Tris, pH 8.0, 500 mM NaCl, 1 mM DTT. The first (and major) peak of the elution profile contained ΔN-CP which was eluted in the exclusion volume, whereas the remaining two peaks contained the MBP and remaining uncleaved fusion protein, respectively. All elution fractions containing ΔN-CP were pooled together, concentrated into a volume of 5 mL and loaded on a pre-equilibrated HiLoad 16/60 Superdex 200 column. The elution profile revealed a single broad peak ranging from 500 to 50 kDa (elution volume from 60 to 80 mL). Fractions containing the purified ΔN-CP were then pooled and stored at 4˚C.

ΔN-D87-D88-CP was loaded on a pre-equilibrated HiLoad 16/60 Superdex 200 column connected with HisTrap FF column. Fractions containing the purified ΔN-D87-D88-CP were then pooled and stored at 4˚C.

## Negative-stain electron microscopy of ΔN-CP and ΔN-D87-D88-CP

Three microlitres of ΔN-CP or ΔN-D87-D88-CP at a final concentration of about 0.05 mg/mL were deposited on glow-discharged carbon-coated copper grid. Excess solution was blotted after 2 min and 4 μL of 1% uranyl acetate was added on the grid for 1 min. The grids were then dried and kept in a desiccator cabinet until observation. Images were recorded with a JEM-1400 Flash, operating at 120 kV, using a Oneview camera (Gatan inc.), at a magnification of X50,000 with applied defocus values ranging from 0.4 to 1.0 μm.

## Figure preparation

Figs 1–7 were prepared using the PyMOL Molecular Graphics System version 2.2.0 (Schrödinger, LLC) and UCSF Chimera (developed by the Resource for Biocomputing, Visualization, and Informatics at the University of California, San Francisco, with support from NIH P41-GM103311) [66].

## Supporting information

**S1 Fig. Cryo-EM characterization of FBNSV viral particles.** (A) A representative cryo-EM image of FBNSV viral particles (scale bar: 50 nm). (B) Central sections of FBNSV 3D classes. The central part of the particles was masked out. The particles contained in the 3D class marked with the red frame were used for the final 3D refinement. (C) FBNSV cryo-EM reconstructed density (EMD-10097). The two isosurfaces (unsharpened and sharpened map) are coloured according to the estimated local resolution. (D) Sectional view (one octant removed) of the unsharpened and sharpened densities. Colours as in C, except for a single CP subunit which is highlighted in white for reference. The position of a genome density residual is indicated by the arrows. (E) Fourier Shell Correlation curve (excerpt from the EMD-10097 validation report).
(TIF)

**S2 Fig. Validation metrics of the refined atomic model of the FBNSV icosahedral capsid (PDB-6S44).** (A) Global validation metrics values, percentile scores (ranging between 0–100), and number of entries on which the scores are based. (B) Excerpt from the wwPDB validation report. The first (top) graphic for the CP chain summarises the proportions of the various outlier classes displayed in the second (bottom) graphic. A dot represents fractions ≤ 5%. The upper red bar indicates the fraction of residues that have poor fit to the EM map (all atom inclusion < 40%). The second graphic shows the sequence view annotated by issues in geometry and atom inclusion in map density. Residues are colour-coded according to the number of geometric quality criteria for which they contain at least one outlier: green = 0, yellow = 1, orange = 2 and red = 3 or more. A red diamond above a residue indicates a poor fit to the EM map for this residue (all atom inclusion < 40%). Stretches of 2 or more consecutive residues without any outlier are shown as a green connector. Residues present in the sample, but not in the model, are shown in grey.
(TIF)

**S3 Fig. FBNSV CP subassemblies around the icosahedral symmetry axes (views from the capsid exterior).** Zones involved in inter-subunit contacts are coloured green and red. The position of Ser 87 and Ser 88, which have been targeted for site directed mutagenesis, is

indicated by the blue spheres.
(TIF)

**S4 Fig. Detection of mutated FBNSV CP S87-R88 in agroinfiltrated plant leaves.** Two leaves per plant were agroinfiltrated as described in Materials and Methods. Two sets of eight plants were respectively infiltrated with wild type (N˚ 1 to 8) and mutated (N˚ 9 to 16) CP clones, together with the seven other segments. One of the two infiltrated leaves of each plant was collected 4 days post-agroinfiltration (A), and the other 6 days post infiltration (B). The CP protein detection is indicated with a small star in each lane. The various leaves expressed the CP protein inconsistently, and we have no explanation for this observation, other than a technical hurdle related to inefficient infiltration. It is also notable that the S87-R88 CP is expressed in a reasonable number of leaves but in amounts seemingly smaller than the wild type. We assume that failure of the mutated CP to assemble virus particle may affect its stability. The molecular weight markers are on the side of the gels and corresponds (from bottom to top) to 15, 25, 35, 55, 70, 100, 130 and 250 kDa. Plant N˚ 9 in A and B was negative and is not shown. Plant N˚ 10 died in between the two sampling dates. In an additional control experiment (C), we also verified that mutation of segment S did not impair its capacity to replicate. Total DNA was extracted from infiltrated leaves, submitted to RCA amplification and digested with AatII, as described in the Materials and Methods. In the agarose gel, the infiltrated segment S or derivative mutants are: line 1 = E/S87-Rev clone 6 (6 dpi), lane 2 = E/S87-Rev clone 8 (6 dpi), lane 3 = S87-R88 clone 10 (4 dpi), lane 4 = S87-R88 clone 11 (4 dpi), lane 5 = S87-R88 clone 16 (4 dpi), lane 6 = S87-R88 clone 10 (6 dpi), lane 7 = S87-R88 clone 15 (6 dpi), lane 8 = mock infiltrated leaf, lane 9 = leaf infiltrated with wild type FBNSV segments but omitting segment R, lane 10 = leaf infiltrated with all 8 wild type segments. Molecular weight marker on right and left is TrackIt 1 kb DNA ladder (Thermofisher Scientific). An RCA amplified band of approximately 1 kb reveals replication of the infiltrated segments (see replication negative control in lane 9). The table on the right shows the qPCR estimate of the copy number of segment S in the RCA product diluted 1/1000. Both the mutated (S87-R88 extracted at 4 dpi) and wild type (E/S87-Rev extracted at 6 dpi) segments accumulate to similar levels in infiltrated leaves.
(JPG)

**S5 Fig. Cryo-EM image processing workflow.**
(TIF)

## Acknowledgments

We are grateful to Sophie Leblaye for all plants/aphids maintenance, production and agroinoculation.

## Author Contributions

**Conceptualization:** Stefano Trapani, Jean-Louis Zeddam, Stéphane Blanc, Patrick Bron.

**Data curation:** Stefano Trapani.

**Formal analysis:** Stefano Trapani.

**Funding acquisition:** Stéphane Blanc, Patrick Bron.

**Investigation:** Stefano Trapani, Eijaz Ahmed Bhat, Michel Yvon, Joséphine Lai-Kee-Him, François Hoh, Marie-Stéphanie Vernerey, Elodie Pirolles, Mélia Bonnamy, Jean-Louis Zeddam, Stéphane Blanc, Patrick Bron.

**Project administration:** Stéphane Blanc, Patrick Bron.

**Resources:** Guy Schoehn.

**Supervision:** Stéphane Blanc, Patrick Bron.

**Validation:** Stefano Trapani, Jean-Louis Zeddam, Stéphane Blanc, Patrick Bron.

**Visualization:** Stefano Trapani.

**Writing – original draft:** Stefano Trapani, Stéphane Blanc.

**Writing – review & editing:** Stefano Trapani, Guy Schoehn, Jean-Louis Zeddam, Stéphane Blanc, Patrick Bron.

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
