## [Decision Letter · Decision Letter 0]

15 Nov 2022

Dear Prof. Trapani,

Thank you very much for submitting your manuscript "Structure-guided mutagenesis of the capsid protein indicates that a nanovirus requires assembled viral particles for systemic infection." for consideration at PLOS Pathogens. As with all papers reviewed by the journal, your manuscript was reviewed by members of the editorial board and by several independent reviewers. In light of the reviews (below this email), we would like to invite the resubmission of a revised version that takes into account the reviewers' comments.

A major issue that needs to be addressed in the revision is regarding virion formation and whether the viral coat protein is necessary for the spread of viral genomic components. Other comments could be addressed by editing the text.

We cannot make any decision about publication until we have seen the revised manuscript and your response to the reviewers' comments. Your revised manuscript is also likely to be sent to reviewers for further evaluation.

Sincerely,

Savithramma P. Dinesh-Kumar

Academic Editor

PLOS Pathogens

Bart Thomma

Section Editor

PLOS Pathogens

Kasturi Haldar

Editor-in-Chief

PLOS Pathogens

orcid.org/0000-0001-5065-158X

Michael Malim

Editor-in-Chief

PLOS Pathogens

orcid.org/0000-0002-7699-2064

Reviewer's Responses to Questions

**Part I - Summary**

Reviewer #1: The manuscript entitled “Structure-guided mutagenesis of the capsid protein indicates that a nanovirus requires assembled viral particles for systemic infection” submitted by Trapani et al. is a very thorough study. The authors determine the structure of FBNSV and then use the structure to cleverly address a significant question regarding the infection and pathogenesis of a very multipartite plant virus. Nanoviruses are neither genetically nor biochemically tractable systems. The authors should be commended for their perseverance and creativity.

Although this was the first time I saw the manuscript, I noted it was a resubmission. The document contained the authors’ responses to four previous reviewers. To remain unbiased by the previous reviews, I first read the manuscript. Afterwards, I read the previous critiques and the authors’ responses.

I had no major concerns regarding the manuscript or the scientific rigor of the research, which I considered high. This is likely due to the comments of the previous reviewers. Below follows some minor points from my reading. After that, my comments regarding the previous reviews and the authors responses.

Minor points:

1. Line 82: Place (M-Rep) at the end of the sentence.

2. Line 92-97: This run-on sentence would best be divided into two for clarity. I suggest, "A study on Faba Bean Necrotic Stunt Virus (FBNSV) [11] showed that a complete set of the eight genomic segments rarely occurs in individual cells of an infected plant. However, the function encoded by a given segment can complement the others at a distance, in cells where this segment is absent.”

3. Line 98: replace either with “e. g.”

4. Line 104: FBNSV, and other multipartite viruses,

5. Line 107-108: or alternatively as assembled viral particles.

6. Line 122-125 I suggest the following rewording: The modified coat proteins could be expressed and accumulated into plant tissues; however, the long distance movement of the virus, and thus systemic infection of the host plant, was abolished in all cases.

7. Line 143: I suggest: This heterogenous particle population, vis-a-vis DNA content, was then investigated by cryo-EM single particle analysis.

8. Line 231-233: subordinate clauses with the word "which" are separated from the main part of the sentence with commas not parentheses. In fact, most of the parentheses in this paragraph should be commas.

9. Legend to Figure S4: “inconsistently” not “unconstantly.”

Comments regarding the previous reviews and the authors’ responses

Firstly, I did not see the earlier version of this manuscript. Thus, my comments here reflect neither the quality of the review process nor the quality of the first submission, in which something may have been poorly explained. I read all of the comments from the last four reviews and the responses. When it was possible to respond positively, the authors made changes. In those instances when a positive response would require conducting experiments that are not feasible in this less than tractable system, the authors should not be penalized. As Otto Von Bismarck stated about politics, research is also the art of the possible. This must be considered when the research system is very intractable, as is the case with nanoviruses.

1. In the previous reviews, some of the reviewers requested experimental validation that the mutant coat proteins could not form capsids. Although the serine residues were very logical targets, this was a reasonable criticism. In this version of the manuscript, the authors provided that data. Thus, should no longer be a concern.

2. In the previous reviews, concern was expressed regarding the low efficiency of the wild-type infections. One reviewer expressed that it should be near 100%. Like the authors, I do not believe that positive controls need approach 100%. They need only set an upper limit and demonstrate how wild-type behaves in the same assay. The numbers given in Table 3 are quite compelling: 187/807 (~23%) of the plants used in the positive control developed symptoms, whereas 8/777 (~1%) of the mutant infections developed symptoms, but the wild-type coat protein gene, due to reversion, was detected in every one of those cases. Thus, it is <1/777 (0.1%). That is more than two orders of magnitude less than the positive control. Despite the low efficiency of the wild-type infections, the authors have rigorously demonstrated, by assaying almost 1600 plants, that the mutant coat proteins do not cause systemic infection. I agree with the authors responses on this point.

2. DNA binding studies: past reviewers stated that the manuscript should include DNA binding studies with wild-type and mutant coat proteins. In light of the in vitro work, which demonstrates that the mutant coat protein only forms pentamer, the results of such studies would be tangential, not altering the authors’ conclusions regarding assembly competent capsids being needed for systemic infection. Moreover, without a clear understanding of nanovirus assembly and DNA packaging, the results of the experiment may be meaningless, or the experiment could not be conducted.

I searched PubMed for manuscripts on Nanovirus or Nanoviridae assembly and packaging, less than 10 manuscripts were identified, and the results of those manuscripts do not address the central question: do capsids assemble around the ssDNA genomes or are ssDNA genomes pumped into empty capsids? If the genome is pumped into a preformed capsid, the DNA interacting domains would be on the interior of the capsid and likely not available for nucleic acid interactions. If pentamers are the assembly subunit, as the data suggests, and is observed in other T=1 ssDNA viruses, one would need full length coat protein pentamers, which must include the positively charged N-terminus. However, the inclusion of this terminus leads to very little expression. Although this paper does contain some intriguing insights into assembly and packaging, the data can be interpreted to weakly support both established models. Moreover, nanovirus assembly is not the topic of this manuscript, and the in vitro assembly system may take years to develop. Thus, I agree with the authors’ responses on this point.

3.Alternate hypotheses: several reviewers put forth alternative hypotheses. While these hypotheses could be occurring, they are far from the simplest hypothesis. For example, one such hypothesis states that the changes to the serine codons are acting on the DNA level, causing the segment to “behave badly.” In this model, 1, 2, or 3 the altered nucleotides would affect a critical DNA structure or sequence substrate required to specifically move mutated DNA segment independently of an assembly competent coat protein. If this were the case, the sequence or a specific ssDNA structure was altered in making the mutations would be conserved in every genomic segment. Moreover, there is no evidence from the structure that the altered region of the coat protein is interacting with nucleic acid. Yes, this hypothesis is possible, but much more complex than the simplest hypothesis that the authors used to interpret their data. The authors do state in their manuscript, “We are perfectly aware that unpredicted effects of these mutations on the CP-DNA interactions cannot be totally excluded.” I think that is sufficient.

4. Structure determination, ordered DNA and the lack of observable heterogeneity. I do agree with the reviewers when they state that the inclusion of more particles may be required to detect heterogeneity. However, considering the 3.2 Å resolution of the current model, these would be very subtle differences and would require a gargantuan effort, especially considering the less than tractable nature of the system. Moreover, greater resolution may not detect any differences. The organization of ssDNA within icosahedral capsids is extremely difficult to study. In most systems, a portion of the nucleic acid backbone is visible as ordered; however, distinct and recognizable nitrogenous bases are rarely, or have yet to be, observed. While not all systems are the same, my group has attempted several techniques including asymmetric reconstructions working with a virus for which purification yields particle concentrations of 1e13/ml. We have yet to see heterogeneity within the populations. We have attempted symmetric reconstructions with viable mutants that alter the path of the DNA during packaging that lead to dramatically difference solution properties. Although this did alter the pattern of the ordered DNA, the protein component of the structures did not differ from wild-type. The dynamics of virions in solution at 20°C are likely not reflected in liquid ethane, or in crystals. Importantly, the resolution of their structure was sufficient to identify mutations that would block particle assembly. I do not believe that analyzing more particles is likely to detect heterogeneity, and if it did, I do not see how it would change the interpretation of the biological assay: systemic infection requires an assembly competent coat protein. Eventually the authors may wish to look for different solution properties, if they can be detected, but developing the assays that could detect these differences will likely take years, especially since they do not have, or will ever have, a simple plaque assay. And lastly, it is not the topic of the manuscript.

Reviewer #2: What does 'several' mean?

Please use 'spread' rather than 'propagate'

Or concentrated in certain cells or tissues?

The last sentence doesn't make sense nor really reflect the results of the study-please delete or re-word

Reviewer #3: The study by Trapani et al., "Structure-guided mutagenesis of the capsid protein suggests that a nanovirus requires formed viral particles for systemic infection," provides novel and intriguing information on FBNSV. The paper reports the structure of the virion using cryo-EM, experimentally confirms the importance of residues Ser 87 and Ser 88 in the formation of dimers, and hypothesises about how systemic spread works. Despite how challenging it is to deal with ssDNA viruses from nanoviruses, the authors have done an admirable job of uncovering their structure and propagation. The work is well done, and most of the conclusions are supported by the findings. All of the significant issues were brought up by the prior reviewers, and the authors have convincingly responded. The only point I want to raise is that the structural work on BBTV CP (ref. 27), which reported similar structural results, needs mention where the authors compare FBNSV with AYVV (pages 19, 20). Yes, a lot of unanswered questions remain, and I am sure the authors will take them up subsequently. However, the present work is no less significant, and it definitely deems publication in Plos Pathogen. I congratulate the authors on their tremendous efforts in comprehending nanovirus capsid assembly.

Reviewer #4: This manuscript describes the determination of the cryoEM structure of the nanovirus, FBNSV, to near-atomic resolution. Though the authors anticipated that they might be able to observe differences in the structures of the particles encapsidating the different DNA components, this proved not to be the case. This is not particularly surprising given that all 8 cssDNA genomic segments are very similar in size (about 1kb) and must be able to interact productively with the coat protein. Nonetheless, the structure obtained is interesting and the methods used for its determination appear to be robust.

In addition to the straightforward cryoEM analysis, the authors also undertake mutagenesis experiments to identify amino acids important for the formation of capsids. In particular, they targeted S87 and S88 which are present at the two-fold axes. Introducing charged amino acids appears to negatively impact virus assembly and abolish or greatly reduce infectivity. In the case of reduced infectivity, revertants are found to accumulate. This is fine and is consistent with the amino acids being important for capsid formation. They then go on to postulate that this indicates that the virus moves in the form assembled particles. This is where I think the manuscript goes a bit too far as I don't think their data actually addresses the proposed models of the mechanism of virus spread (the pluri-cellular way and the need for virus encapsidation). Given this I would rather the authors stick to the structural studies, coupled to mutagenesis of amino acids predicted to be essential for capsid formation, rather than stretch the story to include speculation about the entire replication cycle of the virus.

**Part II – Major Issues: Key Experiments Required for Acceptance**

Reviewer #1: None

Reviewer #2: 1. My major concerns regarded the evidence of virion formation (or not) and this was a concern of the other reviewers. This seems even more important given the mutations are predicted to make virions unstable and this could be impacted in planta. This was partially addressed by the experiment showing E. coli-expressed mutant CP will not form full virions, whereas wild-type CP did. Nonetheless, the evidence still is somewhat circumstantial. Thus, it was disappointing that the ISEM method was so quickly dismissed as‘unimaginable’ (Author response #16). Trying to scale it up would not be that hard. Moreover, the ISEM was effectively used for this purpose, i.e., correlating virion formation with systemic movement, for the monopartite geminivirus BCTV (Soto et al., 2005 Virology). It is fair to say that this approach has been tried, but was it shown there was differential replication in mesophyll vs phloem cells? What about aphid transmission-failure to be aphid-transmitted also would support lack of virion formation.

2. I was puzzled that the authors chose not to use much of the literature of geminivirus long-distance movement. In multiple cases of monopartite geminiviruses evidence for the requirement of virions for systemic infection has been presented (MSV, BCTV), but for some bipartite species, the CP is dispensable. Thus, a non-virion small ssDNA virus can move in non-virion form and cause symptoms. The geminiviruses may also provide another model for long distance movement of nanoviruses: the ‘waves of infection’ of protophloem cells that ultimately give rise to mature sieve elements and their associated companion cells. This infection of the protophloem occurs through the Achilles heel of these progenitor cells developing into sieve element with large sized pores through which virions can pass and released ssDNA that enter the nucleus for replication of formation of more virions. The other way is mature companion cells are infected but various combinations of virions and this could happen via their unique deltoid plasmodesmata with a higher molecular size exclusion limit.

Reviewer #3: None

Reviewer #4: The major comment I have is that if the authors wish to examine whether the viral coat protein is necessary for the spread of viral genomic components, why not simply omit Segment S from the inoculum? If the coat protein essential for spread of the virus, inoculation with just the other 7 segments should not result in infection. Has this been attempted and, if so, what was the result? Making mutations with Segment S seems a much more convoluted way of investigating whether the coat protein is essential, particularly as point mutations are prone to reversion (which is observed).

The other issue with the mutagenesis approach is that aberrant coat protein might have a dominant negative effect on the ability of the virus infection to spread. The formation of aberrant capsids could sequester the cssDNA and inhibit productive infection. A way of showing this not the case is to simply omit Segment S and compare the results with those found when wt or mutant S is included. I really do feel this is an essential experiment.

**Part III – Minor Issues: Editorial and Data Presentation Modifications**

Reviewer #1: (No Response)

Reviewer #2: -I also agreed with reviewer #2 that DNA binding assays with wild-type and mutant proteins would be useful, as would showing replication of the mutant DNAs compared with wild-type.

-Regarding the infectivity data, I also share the concerns about low infectivity of the positive control but I agree with the authors that it is a very challenging experimental system and they address that with numbers of plants and replications. Regarding author response #7, it so happens agroinoculation of geminiviruses at least into common bean, cucurbit, tobacco and tomato routinely gives 100% infection and the key it delivery to the area of those protophloem cells just beneath the shoot apex, so it is possible.

Of course nanoviruses have many more components and I agree that making a multimeric binary vector with all of them has been tried. Not sure about have 2-3/binary as it is possible to have at least a begomovirus and betasatellite on the same binary.

I agree with the authors on the silent mutation experiment, but I think the replication of the mutants, e.g., in a leaf disc assay should be demonstrated.

Reviewer #3: The structural work on BBTV CP (ref. 27), which reported similar structural results, needs mention where the authors compare FBNSV with AYVV (pages 19, 20).

Reviewer #4: On a more minor issue, I felt that much of the material in the first paragraph of the "Discussion" would be better positioned in the "Introduction".

PLOS authors have the option to publish the peer review history of their article (what does this mean?). If published, this will include your full peer review and any attached files.

Reviewer #1: No

Reviewer #2: No

Reviewer #3: No

Reviewer #4: **Yes: **George Peter Lomonossoff
---

## [Editor Report · Decision Letter 1]

27 Dec 2022

Dear Prof. Trapani,

We are pleased to inform you that your manuscript 'Structure-guided mutagenesis of the capsid protein indicates that a nanovirus requires assembled viral particles for systemic infection.' has been provisionally accepted for publication in PLOS Pathogens.

Best regards,

Savithramma P. Dinesh-Kumar

Academic Editor

PLOS Pathogens

Bart Thomma

Section Editor

PLOS Pathogens

Kasturi Haldar

Editor-in-Chief

PLOS Pathogens

orcid.org/0000-0001-5065-158X

Michael Malim

Editor-in-Chief

PLOS Pathogens

orcid.org/0000-0002-7699-2064
---

## [Editor Report · Acceptance letter]

4 Jan 2023

Dear Prof. Trapani,

We are delighted to inform you that your manuscript, "Structure-guided mutagenesis of the capsid protein indicates that a nanovirus requires assembled viral particles for systemic infection.," has been formally accepted for publication in PLOS Pathogens.

Best regards,

Kasturi Haldar

Editor-in-Chief

PLOS Pathogens

orcid.org/0000-0001-5065-158X

Michael Malim

Editor-in-Chief

PLOS Pathogens

orcid.org/0000-0002-7699-2064